# Serving Sizes and Energy Content of Grab-and-Go Sweetened Beverages in Australian Convenience Stores, Supermarkets, and Fast-Food Outlets

**Qingzhou Liu** [1,2], **Jing Ying Lai** [2,3,†], **Kylie Nguyen** [2,3,†] **and Anna Rangan** [2,3,*]

1   School of Life and Environmental Sciences, Faculty of Science, The University of Sydney, Sydney, NSW 2006, Australia; qingzhou.liu@sydney.edu.au
2   Charles Perkins Centre, The University of Sydney, Sydney, NSW 2006, Australia
3   Discipline of Nutrition and Dietetics, Susan Wakil School of Nursing and Midwifery, Faculty of Medicine and Health, The University of Sydney, Sydney, NSW 2006, Australia
*   Correspondence: anna.rangan@sydney.edu.au; Tel.: +61-2-9351-8316
†   These authors contributed equally to this work.

**Abstract:** There is a growing demand for convenience grab-and-go (GNG) food and beverages due to the modern, busy lifestyle. The types of food and beverages available in this sector are currently not well defined, although a large selection of discretionary foods is typically on display. The aims of this study were (1) to summarise the literature on consumers' purchasing behaviours of sweetened beverages, in particular the effects of purchasing locations and settings, price, promotion, and serving sizes, followed by (2) a cross-sectional audit of available sweetened beverages (sugar-sweetened and intensely sweetened) in the GNG sector. Three common GNG settings (convenience stores, front of supermarket, and fast-food outlets) within metropolitan Sydney, Australia, were selected in three different socioeconomic localities. Data were collected by in-store visits (n = 18) and using brand websites between March and April 2022. A total of 1204 GNG sweetened beverages were included. Sugar-sweetened beverages comprised 67% of beverages, with the highest proportion in fast-food outlets (80%), followed by convenience stores (67%) and supermarkets (61%). The majority (63%) of sugar-sweetened beverages had an energy content higher than 600 kJ and a serving size over 375 mL. Fast-food outlets in particular had the smallest selection of sugar-sweetened beverages less than or equal to 250 mL (5.1%). No differences across socioeconomic localities in the proportion and serving sizes of available sugar-sweetened beverages were observed. These findings show that the majority of GNG sweetened beverages have large serving sizes and high energy content, and opportunities exist to improve this food environment. An increased selection of smaller serving sizes can nudge consumers towards more appropriate serving size selections.

**Keywords:** grab-and-go; non-alcoholic beverages; package sizes; sugar-sweetened beverages

## 1. Introduction

A large selection of sweetened non-alcoholic beverages is readily available in a variety of settings with high consumption levels globally among children and adults [1–5]. In Australia, the latest national nutrition survey showed that one third of the population consumed a median of one can of sugar-sweetened beverages per day [2]. The excessive consumption of such beverages results in high sugar and energy intakes, poor diet quality, and micronutrient inadequacy [6–9]. A robust body of evidence has accumulated linking sugar-sweetened beverages with dental caries [10], weight gain [1], obesity [1], hypertension [11], diabetes [12,13], cardiovascular disease [1], and mortality [4,14,15]. Furthermore, emerging research suggests that replacing sugar-sweetened beverages with intensely sweetened beverages may not be the best alternative, although further long-term studies are needed [1,12,16].

The food environment, defined as the consumer interface with the food system that encompasses the availability, affordability, convenience, and desirability of foods, has a strong influence on peoples purchasing and consumption habits [17–19]. This can facilitate consumption of sweetened beverages by its ready availability, strong marketing, and relatively low pricing [18]. Additionally, sweetened beverages are commonly sold in large serving sizes [20,21], which can lead to unconscious overconsumption [22,23]. A landmark Cochrane meta-analysis revealed that people consistently consumed more foods and drinks when offered larger size servings than smaller servings, termed the portion size effect [20]. This effect was present regardless of age, gender, BMI, hunger, dietary restraint, and dietary disinhibition [20]. Several mechanisms have been proposed to explain this effect, although the research in this area is ongoing [24–26]. One theory is that consumers use the package size as an informative cue in the portion size decision process for the "normal" or "appropriate" amount of food to consume [25,27]. Thus, people would consume less if presented with a smaller but reasonable amount, whereas presenting a larger amount would likely result in mindless eating and higher intake [23,25,27,28].

This perception of normal portion sizes might, in turn, be influenced by individuals' personal and social norms about what constitutes a suitable amount to eat. As large serving sizes are ubiquitous in the current food environment, the perceived norm of what constitutes a portion size has shifted [22,23]. For example, a typical portion size of soft drink (soda) in the US was 6.5 oz (180 mL) prior to the 1950s, 12 oz (340 mL) in the 1960s, 20 oz (570 mL) in the 1990s, and as high as 1 L currently [29]. Additionally, caloric beverages (as opposed to solid foods) elicit a weaker satiety response with little compensation at subsequent meals, leading to higher daily energy intakes [30]. Thus, collectively, a shift towards larger package sizes contributes to higher energy intakes and can increase the risks of adverse health outcomes [22,23].

A relatively new but rapidly expanding food outlet is the 'grab-and-go' (GNG) or 'on-the-go' sector [31,32]. This sector is characterised by its provision of easily accessible food and beverage options to accommodate the needs of consumers to reduce time and effort associated with meal shopping and preparation [33,34]. In Australia, common GNG settings include fast-food outlets, convenience stores, and more recently—within supermarkets—a defined section near the entrance for GNG products [32,34]. Beverages such as carbonated or non-carbonated drinks that are high in added sugar or intense sweeteners [35] comprise a large proportion of products within this sector [36]. Currently, there is little research on consumers' purchasing behaviours of these products according to food retail types [37], particularly the GNG sector, with scant information on the types, serving sizes and energy contents of sweetened beverages available, and whether these differ by socioeconomic area.

Given the key role of the external food environment on purchasing and eating behaviours [20,38], a better understanding of the food environment would help identify potential areas for public health interventions to improve health outcomes. We hypothesised that differences would be observed across socioeconomic localities in the proportions and serving sizes of sugar-sweetened beverages available in the GNG sector. This theory is based on research showing higher levels of sugar-sweetened beverage consumption by people living in lower socioeconomic areas compared with those living in higher socioeconomic areas [39]. Therefore, the aims of this study were (1) to summarise the literature on consumers' purchasing behaviours of sweetened beverages, in particular the effects of purchasing locations and settings, price, promotion, and serving sizes, followed by (2) a cross-sectional audit of available sweetened beverages (sugar-sweetened and intensely sweetened) in the GNG sector across three socioeconomic localities within Sydney.

## 2. Part One: Background Literature Review on Purchasing Behaviours of Sweetened Beverages

A keyword search ("grab and go", "ready to go", "sugar-sweetened beverages", "consumer behaviour", "serving size", "portion size") was conducted in Medline via Ovid, Passport Euromonitor, and IBISWorld as the most relevant health and marketing

databases. The search (from inception to June 2023) yielded 107 research articles and reports. Backward and forward hand searching was also undertaken using relevant studies from the preliminary search and yielding another 18 articles [40–42].

People purchase sweetened beverages for a wide variety of reasons that are both individual and contextual (Table 1). Individual factors include taste preferences, sociodemographic variables, and individual characteristics such as age and sex, whilst contextual factors include cost, type of food outlet, availability, and marketing [41–44]. Demographic factors associated with sweetened beverage consumption have been widely reported in the literature and consistently show that adolescents and young adults, males, and those of lower socioeconomic position are the highest consumers [2,42,45,46]. However, contextual factors involved in purchasing decisions such as purchase location, price, marketing, and serving sizes of sweetened beverages are less well studied [40,44,47], and are the focus of this review.

**Table 1.** An overview of individual and contextual factors affecting sweetened beverages purchasing behaviours.

| | |
|---|---|
| Individual factors | Taste preferences<br>Sociodemographic variables (for example, education, employment, income)<br>Individual characteristics (for example, age, sex) |
| Contextual factors | Purchase location<br>Price<br>Marketing (for example, brand reputation)<br>Serving sizes |

*2.1. Purchase Location*

The locations for purchasing sweetened beverages are diverse and growing including supermarkets, convenience stores, fast-food outlets, vending machines, gas stations, as well as non-food outlets such as pharmacies, office supply, and outdoor stores [48]. For example, a US study found that 19% of sugar-sweetened beverages were sold in non-food retail stores such as office supply, pharmacies, and dollar stores [48]. The vast majority of soft drinks are sold through supermarkets as confirmed by Australian industry reports [49] due to the wide range of brands and flavours available, and at a lower cost than other retailers [50–52].

Few Australian studies have examined the purchasing location of sweetened beverages by demographics. A 2017 phone survey of 891 adults who purchased sugar-sweetened beverages for personal consumption, at least occasionally, found that the most common purchase locations were supermarkets (56%), convenience stores (19%), and food or entertainment venues (17%) [41]. Purchase location was associated with age, sex, and type of beverage purchased; supermarkets purchases were most likely to be soft drinks, purchased by females, whereas convenience stores purchases were more likely to be energy drinks and purchased by males [41]. Socioeconomic disadvantage was not a significant factor in these purchasing decisions.

Another Australian study of 675 young adults used a validated food diary app to record purchase location of sugar-sweetened beverages [53]. The authors found that the majority of the consumed beverages had been purchased at the supermarket (62%), followed by convenience stores (19%) and other (19%). Supermarket purchases were significantly higher for women than men (67 vs. 47%), whilst purchases at convenience stores were more likely by men than women (33 vs. 15%). It was noted that purchases from convenient stores were more likely to be consumed at the point of purchase.

Similarly, in the USA, National Health and Nutrition Examination Survey (NHANES) data show that supermarkets and grocery stores were the single largest source for purchasing sugar-sweetened beverages among adults [54]. Of the total energy consumed from sugar-sweetened beverages, 52% were purchased from supermarkets and grocery

stores, 16% from fast-food restaurants, 11% from convenience stores, 8% from full-service restaurants, 4% from vending machines, and 9% from other sources.

Convenience stores, corner stores, gas stations, and dollar stores have been shown to be key sugar-sweetened beverages purchasing outlets for low-income communities [55]. These small food stores are generally labelled as a detrimental indicator of food healthfulness in studies assessing the relationship between local food environments and diet as they typically stock an abundance of energy-dense nutrient-poor food options [40]. Several studies have shown that such food stores contribute significantly to the urban food environment [40,55,56]. A survey in Minneapolis–St Paul demonstrated that three quarters of customers shopped at small food stores at least weekly, and the majority of sales were sugar-sweetened soft drinks [40]. Madsen reported purchasing behaviours in low-income communities in California, USA, and found that although the majority of participants (41%) purchased most of their sugar-sweetened beverages from grocery stores (including discount, large-chain, and other grocery stores), 28% purchased from corner stores. Participants who had not graduated from high school were likely to purchase most sugar-sweetened beverages at corner stores or discount grocery stores than at large-chain grocery chains. The authors proposed that this was due to the easier accessibility or closer proximity, which is a known driver of store choice, particularly for those who do not own cars [55].

A survey of 147 convenience stores including gas marts, corner stores, pharmacies, and dollar stores in Minneapolis–St Paul examined food and beverage purchase data using intercept surveys from over 3000 adults between 2014 and 2017 [56]. Over half of all purchases included a sweetened beverage. Sugar-sweetened beverage purchases were highest among males, young adults, and those with lower education and income, whereas artificially sweetened beverage purchases were highest among women, those aged 40–59 years, and with higher education and income levels [56].

Purchases at convenience stores are prone to impulse buys, as shown in a US survey by GasBuddy, a crowdsourcing platform for fuel prices and convenience store data [57]. Over half (51%) of respondents reported purchasing a beverage at a gas station store at least once a week (20% reported daily purchases), with the majority of purchases (65%) being unplanned 'impulse buys'. Carbonated soft drinks, water, and energy drinks were among the top beverages purchased. Market reports confirm that convenience stores display their stock prominently to encourage impulse purchases [50]. These studies highlight the importance of understanding and monitoring beverage purchases in all food outlets as a health and equity issue.

### 2.2. Price

Price is one of the most significant factors that influence consumer behaviour [41,58] and price promotions are widely used by retailers to encourage consumers to purchase larger quantities and/or change brands [59,60]. A modelling study from the UK demonstrated that price promotion results in an additional 22% of foods or beverages sold [61].

Price may be particularly important for males, younger people, and those of lower socioeconomic position, as they were more likely to agree that sugar-sweetened beverages were 'cheap' and 'better value than water' compared to other population subgroups [41]. Young adults and males were also more persuaded by 'meal deals' as a reason to purchase sugar-sweetened beverages [41]. Value-for-money (i.e., resulting in per-unit savings) is a strong motivator for purchasing large or supersized products, and tends to decrease the focus on health importance, especially in the short term [62].

### 2.3. Marketing

Brand reputation is another factor that influences purchasing behaviour, as many consumers are highly brand loyal, particularly for cola-flavoured soft drinks and energy drinks [52,63]. Consequently, these consumers are likely to pay more for their preferred product [63]. Energy drinks containing active ingredients such as caffeine, guarana, or

taurine are marketed as being able to increase energy and improve concentration, with some consumers using them as a substitute for coffee [64]. They are promoted by large marketing campaigns, targeted to adolescents and young adults [52,65,66]. Taste, brand loyalty, and the perceived positive effects of energy drinks have been identified as key drivers of purchasing behaviour and consumption choices [67]. The influence of socioeconomic position on energy drink consumption is still unclear, even though young males are the highest consumers of energy drinks [67,68]. Industry reports indicate that competition in this market is high; thus, manufacturers are responding by increasing packaging sizes to provide better value for money and incentivise consumers [52].

### *2.4. Serving Sizes*

A wide range of package sizes of sweetened beverages are available for purchase [69,70]. An Australian study found that the majority of soft drink purchases in convenience stores were between 400 and 800 mL [41]. Sales data of carbonated beverages between 2005 and 2019 showed an increased growth of smaller package sizes (<300 mL) and a decrease in sales of larger package sizes ($\geq$2000 mL) in the USA, the UK, Canada, and Australia [69]. The key reasons for this trend, identified by food industry, related to consumer health awareness, portion size control, convenience, and government or industry initiatives. Although this trend towards smaller sizes may have changed over the past few years due to the COVID pandemic and stockpiling of foods [71,72], a recent 2022 Euromonitor report predicts that smaller-sized bottles and cans, and affordable multipacks will be introduced in response to the global challenges of inflation [73].

### *2.5. Future Direction*

Industry reports suggest that consumers are more health-conscious now than before the pandemic, and moving away from artificial ingredients and sweeteners towards more functional ingredients [63,74,75].

Manufacturers are expanding their ranges, introducing new products, product innovation, new flavours, and low sugar varieties [52]. Additionally, the growth of online retailers such as Amazon and eBay could broaden the reach of sweetened beverages, and offer a wider range of brands and flavours than found in supermarkets [50].

In conclusion, purchase location, cost, and the marketing and promotion of sweetened beverages can influence the purchasing behaviours of population subgroups. As the food retail environments differ greatly between countries, the findings from one cannot easily be generalised to others. Little research has been undertaken specifically on the GNG environment including the GNG sections within supermarkets.

## 3. Part Two: Cross-Sectional Audit Study

### *3.1. Methodology*

A cross-sectional audit was undertaken to examine and compare the serving sizes and energy contents of GNG sweetened beverages in three common retail settings and across three different localities within Sydney.

#### 3.1.1. Inclusion Criteria

GNG sweetened beverages in the current study are described as sweetened water-based discretionary beverages containing added sugar or intense sweeteners [35], including soft drinks, energy drinks, frozen drinks (for example, slushies and spiders), iced tea, and sugar-sweetened specialty drinks. Such beverages do not fit into the five food groups and are not necessary for a healthy diet as per the Australian Dietary Guidelines [35]. Flavoured milks and fruit juices were excluded, as they are not considered discretionary choices [35].

#### 3.1.2. GNG Settings

Three common GNG settings were investigated, including convenience stores, the 'ready-to-eat' or 'grab-and-go sections' of supermarkets (excluding main aisles), and fast-

food outlets (over the counter purchases). Convenience stores are rapidly expanding in Australia [76], and the main chains located within the three Sydney localities were selected: 7-Eleven, Coles Express, Woolworths Metro, IGA Xpress [76]. The two biggest supermarket chains, Coles and Woolworths with 28% and 37% market share, were selected, as they commonly have a 'grab and go' section [77]. The selected fast-food outlets included McDonald's, KFC, Hungry Jacks, and Oporto [78].

Three localities within Sydney were selected to assess differences in product availability: Sydney Central Business District (CBD), Eastern Sydney, and Western Sydney) due to different consumer demographics. Consumers purchasing in the Sydney CBD include office workers and tourists, while Eastern Sydney has a higher socioeconomic profile (using the Index of Relative Socioeconomic Advantage and Disadvantage, IRSAD) than Western Sydney based on households with higher incomes, more education, larger homes, and more people employed as managers or professionals [39]. Two supermarkets and four convenience stores were chosen in each locality (total of 18 stores).

For fast-food outlets, specific locality data were not collected, as the products were consistent between Sydney outlets.

### 3.1.3. Data Collection

Data on GNG beverage type, serving size, and energy content were collected in-store by two researchers (J.Y.L. and K.N.) using smartphone cameras to capture images of the front and back of packaging. Convenience stores and supermarkets were visited up to three times each to collect a comprehensive range of beverages; if beverages were still unavailable, images of shelf labels were taken for reference and product data were sourced from brand websites or supermarket shelves.

For GNG beverages available in fast-food outlet settings, information on beverage type, serving size, and energy content was obtained by searching the outlets' websites. All data collected in this study were publicly available in-store or online; thus, no ethics approval was required.

### 3.1.4. Data Charting, Cleaning and Categorisation

Collected data including detailed product description (brand, product name), retail settings (convenience store, GNG section of supermarket or fast-food outlet), name of store, locality (for supermarket and convenience store products only), serving sizes (mL), and energy per serving (kJ) information were charted into a pre-designed spreadsheet.

All data were cross-checked by two researchers (J.Y.L. and K.N.) by removing duplicates, finding missing values, and checking for outliers.

Beverages were classified as sugar-sweetened or intensely sweetened (diet)—defined as drinks that contained no added sugars (no honey, malt, or malt extracts) or had a sugar content less than 2.5 g per 100 mL [79]. These were further grouped into five broad categories according to product characteristics: energy drinks, soft drinks, frozen drinks, iced tea, and specialty drinks. The serving sizes were categorised as small ($\leq$250 mL), medium (251–375 mL), or large (>375 mL) based on Australia's Healthy Food Partnership Industry Guide to Voluntary Serving Size Reduction [80].

### 3.1.5. Data Analysis

Descriptive analyses on serving sizes and energy per serving of beverage categories were undertaken. Energy per serving was compared with the reference amount of one discretionary serve (600 kJ) for all beverage categories [35,70,81].

Non-parametric tests were used to test differences between categories due to data constituting ordinal values. The chi-square test, Mann–Whitney-U test and Kruskal–Wallis-H tests were performed to investigate the differences in availability of the types of beverages (sugar-sweetened vs. diet), serving sizes, and energy per serving across the GNG settings and three Sydney localities. Data were analysed using IBM SPSS v28 (IBM, Armonk, NY, USA, 2021) with significant difference set at $p < 0.05$.

### 3.2. Results

The serving sizes and energy information of 1204 GNG beverages were collected and grouped according to the product type (Table 2). The vast majority of GNG beverages were soft drinks (n = 533) and energy drinks (n = 514), followed by frozen drinks (n = 109), iced tea (n = 42), and specialty drinks (n = 6).

**Table 2.** Comparison of serving size and energy content by beverage category.

|  |  | n (%) | Serving Size, Median (Range), mL | Energy Per Serving, Median (Range), kJ |
|---|---|---|---|---|
| Total | Sugar-sweetened | 812 (67.4) | 473 (100–1150) | 709 (168–1505) |
|  | Diet [a] | 392 (32.6) | 473 (237–1150) | 27 (3–177) |
| Soft drinks | Sugar-sweetened | 336 (63.0) | 375 [b] (100–669) | 694 (168–1352) |
|  | Diet | 197 (37.0) | 500 (237–900) | 10 (3–128) |
| Energy drinks | Sugar-sweetened | 339 (66.0) | 440 (250–600) | 854 (195–1470) |
|  | Diet | 175 (34.0) | 473 (250–600) | 43 (6–95) |
| Frozen drinks | Sugar-sweetened | 90 (82.6) | 650 (375–1150) | 780 (251–1505) |
|  | Diet | 19 (17.4) | 591 (340–1150) | 77 (30–177) |
| Iced tea | Sugar-sweetened | 41 (97.6) | 500 (250–500) | 410 (287–465) |
|  | Diet | 1 (2.1) | 450 | 27 |
| Specialty drinks | Sugar-sweetened | 6 (100) | 300 (300–320) | 607 (423–815) |

[a] Diet beverages are described as intensely-sweetened beverages that contained no added sugars (no honey, malt, or malt extracts) or had a sugar content less than 2.5 g per 100 mL [79]. [b] Mann–Whitney-U test conducted, a significant difference in serving sizes was observed between sugar-sweetened and diet soft drinks, $p = 0.025$.

Sugar-sweetened beverages accounted for 67.4% of all GNG beverages available—soft drinks 63.0%, energy drinks 66.0%, frozen drinks 82.6%, iced tea 97.6%, and specialty drinks 100%.

The median serving sizes of sugar-sweetened and diet beverages were similar overall at 473 mL but serving sizes of sugar-sweetened soft drinks were smaller than their diet counterparts (375 mL vs. 500 mL, $p = 0.025$). The selection of serving sizes available for all GNG beverages ranged from 100 mL (mini bottles) to 1150 mL (slushies).

3.2.1. Differences by Retail Settings

A comparison of GNG sugar-sweetened versus diet beverages available in the three retail settings is shown in Table 3. The percentage of sugar-sweetened beverages was the highest in fast-food outlets (79.5%) with only few diet beverage options available (n = 25, 20.5%), followed by convenience stores (67.3%) and supermarkets (60.6%). No significant difference between serving sizes of sugar-sweetened and diet beverages was observed overall or in each setting ($p > 0.05$).

**Table 3.** The serving sizes (mL) and energy per serving (kJ) of sugar-sweetened and diet GNG beverages in three retail settings.

| Retail Settings | Sugar-Sweetened Beverages | | | Diet Beverages | | |
| | % (n) | Serving Sizes [a], mL | | % (n) | Serving Sizes [b], mL | |
| | | Median | Range | | Median | Range |
| --- | --- | --- | --- | --- | --- | --- |
| Convenience stores | 67.3 (592) | 380 | 100–1150 | 32.7 (287) | 473 | 250–1150 |
| Supermarkets | 60.6 (123) | 500 | 250–600 | 39.4 (80) | 600 | 250–600 |
| Fast-food outlets | 79.5 (97) | 473 | 229–669 | 20.5 (25) | 473 | 237–900 |

[a] Kruskal–Wallis-H test conducted. A significant difference of sugar-sweetened beverage serving sizes between retail settings was observed, *p* = 0.006. [b] Kruskal–Wallis-H test conducted. A significant difference of diet beverage serving sizes between retail settings was observed, *p* = 0.038.

A significant difference in serving sizes were noted across settings (*p* = 0.006); those from supermarkets were found to have the largest median serving size (500 mL for sugar-sweetened beverages, 600 mL for diet beverages), followed by fast-food outlets (473 mL for sugar-sweetened and diet beverage) and convenience stores (380 mL for sugar-sweetened and 473 for diet beverages). For both sugar-sweetened and diet beverages, convenience stores had the widest selection of serving sizes available.

The percentages of small, medium, and large sugar-sweetened GNG beverages in three settings are presented in Figure 1. Beverages were classified as small if serving size ≤250 mL, medium if 251–375 mL, large if >375 mL; one standard discretionary serve equals 600 kJ as per the Australian Dietary Guidelines [35,70,81].

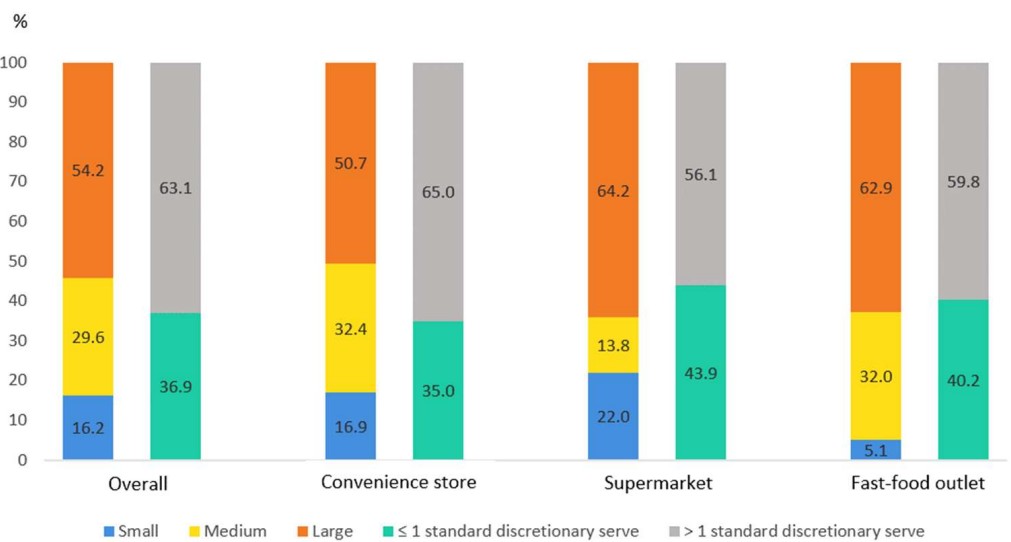

**Figure 1.** The serving sizes and energy content of sugar-sweetened GNG beverages compared to a standard discretionary serve, by setting. Beverages classified as small if serving size ≤ 250 mL, medium if 251–375 mL, large if >375 mL; one standard discretionary serve equals 600 kJ [35,70,81].

Overall, 16.2% of sugar-sweetened beverages had a serving size equal to or smaller than 250 mL, 45.8% of these beverages were equal to or less than 375 mL. The energy content of 36.8% of sugar-sweetened beverages was below or equal to 600 kJ, equivalent to one standard serve for discretionary foods.

The differences in serving sizes available in different GNG settings were noted: 16.9% of sugar-sweetened beverages from convenience stores and 22.0% from supermarkets had a small serving size, compared with a much smaller proportion from fast-food outlets (5.1%). Consistent across all settings, large serving sizes were most commonly available (35.8–49.3%), and the majority of sugar-sweetened beverages contained more than 600 kJ per serving (56.1–65.0%).

### 3.2.2. Differences between Socioeconomic Localities

A comparison of GNG sugar-sweetened versus diet beverages available in three localities showed that the proportion of sugar-sweetened to diet beverage options were similar (Table 4). In addition, no significant differences were found between localities in terms of serving sizes available nor energy content ($p > 0.05$).

**Table 4.** The serving sizes (mL) of sugar-sweetened and diet GNG beverages in supermarkets and convenience stores across three Sydney localities.

| | Sugar-Sweetened Beverages | | | Diet Beverages | | |
|---|---|---|---|---|---|---|
| Localities | % (n) | Serving Size, mL | | % (n) | Serving Size, mL | |
| | | Median | Range | | Median | Range |
| Sydney CBD | 67.0 (309) [a] | 380 | 100–1150 | 33.0 (152) | 487 | 250–1150 |
| Eastern Sydney | 61.8 (175) | 473 | 200–1150 | 38.2 (108) | 500 | 250–1150 |
| Western Sydney | 68.3 (231) | 473 | 100–1150 | 31.7 (107) | 473 | 250–1150 |

[a] Chi-square test conducted across localities, no significant difference in proportions of sugar-sweetened beverages observed ($p = 0.092$).

### 3.3. Discussion

This study provides a novel insight into the variety of sweetened beverages available in the GNG food environment across three socioeconomic localities within Sydney. Over 1200 sweetened beverages were included, consisting of a wide range of serving sizes and energy values. The majority of all GNG beverages were sugar-sweetened and approximately two-thirds exceeded the recommended medium serving size of 375 mL set by the Healthy Food Partnership in The Industry Guide to Voluntary Serving Size Reduction [80]. No differences were observed between the socioeconomic localities in terms of proportion of sugar-sweetened beverages, serving sizes, or energy values.

Energy drinks comprised almost half of GNG sweetened beverage options. Such drinks are frequently marketed as 'functional drinks' that can improve energy and concentration [82], and are becoming increasingly popular especially among young adults [82,83]. However, there is emerging evidence showing that the overconsumption of energy drinks is associated with negative health consequences due to the high sugar and caffeine content [64,82–84].

Our findings highlight the need to actively engage the food industry to provide more appropriate drink serving sizes for this emerging sector, especially for single-serve GNG sweetened beverages that are likely to be consumed by one person in one sitting. The accumulated evidence shows that the high consumption of sugar-sweetened beverages can result in excessive intakes of energy and contribute to obesity, diabetes, cardiovascular disease, and other chronic conditions [1,11,13]. Multiple public health strategies have been proposed to reduce sugar-sweetened beverage consumption such as taxation, banning sales in schools, restrictions on marketing to children, public health education campaigns, front-of-package warning labels, and portion size caps [85–88]. For example, a package cap at 375 mL has been considered a cost-effective approach, with statistical modelling showing the introduction of such a cap is likely to reduce energy intake from sugar-sweetened beverages and resultant weight reductions across the entire population [89]. The Industry Guide to Voluntary Serving Size Reduction, developed by the Healthy Food Partnership in Australia, emphasises the critical role of food industry to introduce smaller drink options to facilitate consumers in making more appropriate serving size choices [80,90,91]. As the acceptability of population-level portion control interventions by consumers and food industry is unclear, careful consideration is required [89,92]. Promisingly, a recent study noted that industry was motivated to increase the availability of smaller options in response to consumers' demand for convenience and increased health consciousness [69].

No differences in available GNG sweetened beverages were noted between the three socioeconomic localities, with the types, serving sizes, and energy content being similar. We acknowledge that the current study included localities within metropolitan Sydney only, although one study conducted in rural Australia noted limited healthier food and drink options [93]. As studies have suggested regional difference and socioeconomic position are predictors of sugar-sweetened beverage consumption [42,94], future research could investigate and compare GNG beverage variety and availability across regions to increase our understanding of food environments in different local contexts.

Strengths and Limitation

Several strengths of this study can be noted. A wide variety of GNG outlets were selected including convenience stores, supermarkets, and fast-food outlets in Sydney, and different localities were included to examine the variation in availability of sweetened discretionary beverages. All data were cross-checked by two independent researchers (K.N. and J.L.) to ensure consistency. However, we acknowledge that limitations exist. This study focused on the availability data relevant to serving sizes and energy content only, the price and value-for-money of GNG beverages, as well as the patterns of sales and consumption, were not investigated. We included sweetened beverages at the most common GNG settings only. Additional GNG beverage types (for example, flavoured milks, mineral drinks, water) and smaller retail outlets (for example, milk bars and corner stores) could be investigated to provide a more detailed scenario. Future studies could consider a wider range of discretionary foods and beverages in the GNG sector, including small-to-medium-sized independent food outlets, and investigate the sales and consumption trends of these products.

## 4. Conclusions and Implication

This study provided a novel insight into the availability of sweetened discretionary drinks in the relatively new GNG food environment in Australia. The majority of GNG beverages had serving sizes exceeding one standard discretionary serve (600 kJ), resulting from large serving sizes > 375 mL, making such sizes the 'norm' in the GNG section. These findings highlight the need to create food environments that enable consumers to choose more appropriate portion sizes, compatible with their energy needs. Active engagement of stakeholders including the food industry, retailers, and government bodies, together with clear guidelines around recommended serving sizes, is critical to assist consumers to select appropriate serving sizes and avoid overconsumption.

**Author Contributions:** Conceptualization, A.R.; data collection and cleaning, J.Y.L. and K.N.; formal analysis, Q.L. and A.R.; writing—original draft preparation, Q.L.; writing—review and editing, Q.L. and A.R. All authors have read and agreed to the published version of the manuscript.

**Funding:** This research received no external funding.

**Data Availability Statement:** Data presented in this study are available on request from the corresponding author.

**Conflicts of Interest:** The authors declare no conflict of interest.

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
