# Peer review of "Serving Sizes and Energy Content of Grab-and-Go Sweetened Beverages in Australian Convenience Stores, Supermarkets, and Fast-Food Outlets"

_beverages, doi:10.3390/beverages9030077_

Round 1

Reviewer 1 Report (Previous Reviewer 2)

The authors addressed all relevant observations made by this reviewer, thus improving paper overall quality. Under these circumstances, there is no more objection to publication by this reviewer.

Author Response

The authors addressed all relevant observations made by this reviewer, thus improving paper overall quality. Under these circumstances, there is no more objection to publication by this reviewer.

Author response: Thank you for your comments and review of this manuscript.

Reviewer 2 Report (New Reviewer)

Dear Authors -

I found this manuscript very well written, addressing a topic in an appropriate scientific manner. I am suggesting the below before the publication is issues:

1 - The title gives the connotation of a much broader study, however, it centers on a specific city retail situation. You may want to consider a new title like "Serving sizes and energy content of grab-and-go sweetened beverage in Australian Urban retail stores" or in an "urban food environment" or else...I think "urban" and/or Australia helps with guiding the reader what to expect. You may find other 

2. alphabetical order is needed in keywords (and perhaps add some that you may remove from the original title)

3. Define "food environment" somewhere as many scientist are not so familiar with it

4. In summarizing the literature, it seems that a table could have been helpful to better position some of the numbers. I mentioned this because if you are indicating this is "a summary of literature" why some of the headlines only have a handful of references?

5.  The survey includes drinks in containers larger than 1 Liter, are those marketed as individual drinks? or is targeting parties of two or so? 

6. Is there is any correlation of price per mL with respect to the size? please consider at least for the discussion

7. The conclusion is a section that needs some work. This section should be dedicated to what the authors strictly found...but you are brining new information, bring some recommendations on what to do (without any reference). Either you move what is not clear conclusion of the study, to another section such as "way forward" or so, or you change the title (for conclusions and final remarks or so)

Author Response

Reviewer 2

Dear Authors -

I found this manuscript very well written, addressing a topic in an appropriate scientific manner. I am suggesting the below before the publication is issues:

Author response: Thank you for your comments and review of this manuscript.

1 - The title gives the connotation of a much broader study, however, it centers on a specific city retail situation. You may want to consider a new title like "Serving sizes and energy content of grab-and-go sweetened beverage in Australian Urban retail stores" or in an "urban food environment" or else...I think "urban" and/or Australia helps with guiding the reader what to expect. You may find other 

Author response: The title has now been amended, “Serving sizes and energy content of grab-and-go sweetened beverages in Australian convenience stores, supermarkets, and fast-food outlets”.

  1. alphabetical order is needed in keywords (and perhaps add some that you may remove from the original title)

Author response: This has now been amended, “grab-and-go, non-alcoholic beverages, package sizes, sugar-sweetened beverages”. (Line 34).

  1. Define "food environment" somewhere as many scientist are not so familiar with it

Author response: The definition of food environment has been added, “The food environment, defined as the consumer interface with the food system that encompasses the availability, affordability, convenience, and desirability of foods, has a strong influence on peoples purchasing and consumption habits [1-3]. This can facilitate consumption of sweetened beverages by its ready availability, strong marketing and relatively low pricing [2]. (Lines 48-52).

  1. In summarizing the literature, it seems that a table could have been helpful to better position some of the numbers. I mentioned this because if you are indicating this is "a summary of literature" why some of the headlines only have a handful of references?

Author response: Thank you for your suggestion. A summary table of individual and contextual factors on sweetened beverages purchasing behaviours has been added for a better overview (Lines 117-120).

  1. The survey includes drinks in containers larger than 1 Liter, are those marketed as individual drinks? or is targeting parties of two or so? 

Author response: The over 1L sizes were found in convenience stores and marketed as individual drinks (for example, Slushies from 7-11 stores).

This information has now been added in the Results section for better clarity, “For both sugar-sweetened and diet beverages, convenience stores had the widest selection of serving sizes available.” (Lines 337-338).

  1. Is there is any correlation of price per mL with respect to the size? please consider at least for the discussion

Author response:

As we focused on the availability data relevant to serving sizes and energy content only, the price and value-for-money were out of scope for this study. This has been acknowledged as a limitation in the Strength and Limitation section:

“However, we acknowledge limitations exist. This study focused on the availability data relevant to serving sizes and energy content only, the price and value-for-money of GNG beverages, as well as patterns of sales and consumption were not investigated. We included sweetened beverages at the most common GNG settings only. Additional GNG beverage types (for example, flavoured milks, mineral drinks, water) and smaller retail outlets (for example, milk bars and corner stores) could be investigated to provide a more detailed scenario.” (Lines 426-432).

Additionally, price has been included in the Literature review section as one of contextual factors influencing sweetened beverages purchasing behaviours:

“Price is one of the most significant factors that influence consumer behaviour [4,5] and price promotions are widely used by retailers to encourage consumers to purchase larger quantities and/or change brands [6,7]. A modelling study from the UK demonstrated that price promotion results in an additional 22% of foods or beverages sold [8].

Price may be particularly important for males, younger people and those of lower socioeconomic position as they were more likely to agree that sugar-sweetened beverages were ‘cheap’ and ‘better value than water’ compared to other population subgroups [4]. Young adults and males were also more persuaded by ‘meal deals’ as a reason to purchase sugar-sweetened beverages [4]. Value-for-money (i.e. resulting in per-unit savings) is a strong motivator for purchasing large or supersized products, and tends to decrease the focus on health importance, especially in the short term [9].” (Lines 186-196).

  1. The conclusion is a section that needs some work. This section should be dedicated to what the authors strictly found...but you are brining new information, bring some recommendations on what to do (without any reference). Either you move what is not clear conclusion of the study, to another section such as "way forward" or so, or you change the title (for conclusions and final remarks or so)

Author response:   The heading of this section has now been amended as ‘Conclusions and implication” (Line 439).

This section has been amended as per suggested, “This study provides a novel insight into the availability of sweetened discretionary drinks in the relatively new GNG food environment in Australia. The majority of GNG beverages had serving sizes exceeding one standard discretionary serve (600 kJ), resulting from large serving sizes >375 mL making such sizes the ‘norm’ in the GNG section. These findings highlight the need to create food environments that enable consumers to choose more appropriate portion sizes, compatible with their energy needs. Active engagement of stakeholders including the food industry, retailers and government bodies, together with clear guidelines around recommended serving sizes, is critical to assist consumers to select appropriate serving sizes and avoid overconsumption.”(Lines 437-445).

References

  1. Downs, S.M.; Ahmed, S.; Fanzo, J.; Herforth, A. Food Environment Typology: Advancing an Expanded Definition, Framework, and Methodological Approach for Improved Characterization of Wild, Cultivated, and Built Food Environments toward Sustainable Diets. Foods 2020, 9, doi:10.3390/foods9040532.
  2. Gamba, R.J.; Schuchter, J.; Rutt, C.; Seto, E.Y. Measuring the food environment and its effects on obesity in the United States: a systematic review of methods and results. J Community Health 2015, 40, 464-475, doi:10.1007/s10900-014-9958-z.
  3. Herforth, A.; Ahmed, S. The food environment, its effects on dietary consumption, and potential for measurement within agriculture-nutrition interventions. Food Security 2015, 7, 505-520, doi:10.1007/s12571-015-0455-8.
  4. Dono, J.; Ettridge, K.; Wakefield, M.; Pettigrew, S.; Coveney, J.; Roder, D.; Durkin, S.; Wittert, G.; Martin, J.; Miller, C. Nothing beats taste or convenience: a national survey of where and why people buy sugary drinks in Australia. Aust N Z J Public Health 2020, 44, 291-294, doi:10.1111/1753-6405.13000.
  5. Nguyen, H. Consumption of several types of soft drinks remained stable year-over-year in Great Britain. Availabe online: https://yougov.co.uk/topics/consumer/articles-reports/2021/02/18/soft-drinks-gb-research-2020 (accessed on 29 June 2023).
  6. Bennett, R.; Zorbas, C.; Huse, O.; Peeters, A.; Cameron, A.J.; Sacks, G.; Backholer, K. Prevalence of healthy and unhealthy food and beverage price promotions and their potential influence on shopper purchasing behaviour: A systematic review of the literature. Obesity reviews 2020, 21, e12948-n/a, doi:10.1111/obr.12948.
  7. Zorbas, C.; Gilham, B.; Boelsen-Robinson, T.; Blake, M.R.C.; Peeters, A.; Cameron, A.J.; Wu, J.H.Y.; Backholer, K. The frequency and magnitude of price-promoted beverages available for sale in Australian supermarkets. Aust N Z J Public Health 2019, 43, 346-351, doi:10.1111/1753-6405.12899.
  8. Smithson, M.; Kirk, J.; Capelin, C. An analysis of the role of price promotions on the household purchases of food and drinks high in sugar, and purchases of food and drinks for out of home consumption. Availabe online: https://assets.publishing.service.gov.uk/government/uploads/system/uploads/attachment_data/file/947412/Sugar_Reduction_analysis_of_price_promotions_on_the_household_purchases_of_food_and_drinks_high_in_sugar__4_.pdf (accessed on 26 June 2023).
  9. Haws, K.L.; Winterich, K.P. When Value Trumps Health in a Supersized World. J Mark 2013, 77, 48-64.

Reviewer 3 Report (New Reviewer)

The paper addresses an important topic concerning the convenience and attractivity of consumption of sweetened beverages. I have just some minor comments to the manuscript.

Firstly, as readers can be expected to come from all over the world, and food outlets differ from country to country, the authors must mention where the study is conducted.

Abstract: Include where the study is conducted.

Introduction:

Line 78: In Australia, common GNG settings…

L (91-)93-97: Move and rewrite. The background for this hypothesis should be addressed and expanded upon earlier in the chapter.

Aims L 86-90: You might consider adding why you are doing this (lines 72-74), not only what you aim to do.

Structure of the headings: Consider changing the headings into study 1 and 2, because it is very strange to read “2. Background”, followed by “3. Methods”, when these are two separate studies.

Lines 99-103: Please include more details on the results of the literature search.

L 123-125: Please write in the text that this applies to Australia.

Discussion:

The first paragraph is actually a summary of the results, consider moving or rephrasing this paragraph.

Conclusions:

L439-440: Add “in Australia”.

L444-447- (450): Move to Discussion. This has not been discussed previously, and new information should not be presented in Conclusions. Also, references should not be included in Conclusions.

Author Response

Reviewer 3

The paper addresses an important topic concerning the convenience and attractivity of consumption of sweetened beverages. I have just some minor comments to the manuscript.

Author response: Thank you for your comments and review of this manuscript.

Firstly, as readers can be expected to come from all over the world, and food outlets differ from country to country, the authors must mention where the study is conducted.

Author response: The title has been amended as “Serving sizes and energy content of grab-and-go sweetened beverages in Australian convenience stores, supermarkets, and fast-food outlets”.

Abstract: Include where the study is conducted.

Author response: This information has now been added, “Three common GNG settings (convenience stores, front of supermarket, and fast-food outlets) within metropolitan Sydney, Australia, were selected in three different socioeconomic localities.” (Lines 21-23)

Introduction:

Line 78: In Australia, common GNG settings…

Author response: This has been amended, “In Australia, common GNG settings include fast food outlets, convenience stores, and more recently, within supermarkets – a defined section near the entrance for GNG products”. (Lines 76-77).

L (91-)93-97: Move and rewrite. The background for this hypothesis should be addressed and expanded upon earlier in the chapter.

Author response: This section has been amended as per suggestions,

“Given the key role of the external food environment on purchasing and eating behaviours [1,2], a better understanding of the food environment would help identify potential areas for public health interventions to improve health outcomes. We hypothesised that differences would be observed across socioeconomic localities in the proportions and serving sizes of sugar-sweetened beverages available in the GNG sector. This theory is based on research showing higher levels of sugar-sweetened beverage consumption by people living in lower socioeconomic areas compared with those living in higher socioeconomic areas [3]. Therefore, the aims of this study were 1) to summarise the literature on consumers’ purchasing behaviours of sweetened beverages, in particular the effects of purchasing locations and settings, price, promotion, and serving sizes, followed by 2) a cross-sectional audit of available sweetened beverages (sugar-sweetened and intensely-sweetened) in the GNG sector across three socioeconomic localities within Sydney.” (Lines 85-96).

Aims L 86-90: You might consider adding why you are doing this (lines 72-74), not only what you aim to do.

Author response: This section has been amended to better explain the scope of this study,

““Given the key role of the external food environment on purchasing and eating behaviours [1,2], a better understanding of the food environment would help identify potential areas for public health interventions to improve health outcomes. We hypothesised that differences would be observed across socioeconomic localities in the proportions and serving sizes of sugar-sweetened beverages available in the GNG sector. This theory is based on research showing higher levels of sugar-sweetened beverage consumption by people living in lower socioeconomic areas compared with those living in higher socioeconomic areas [3]. Therefore, the aims of this study were 1) to summarise the literature on consumers’ purchasing behaviours of sweetened beverages, in particular the effects of purchasing locations and settings, price, promotion, and serving sizes, followed by 2) a cross-sectional audit of available sweetened beverages (sugar-sweetened and intensely-sweetened) in the GNG sector across three socioeconomic localities within Sydney.” (Lines 85-96).

Structure of the headings: Consider changing the headings into study 1 and 2, because it is very strange to read “2. Background”, followed by “3. Methods”, when these are two separate studies.

Author response: We have amended the headings and numberings; subheadings were used throughout this section to  improve the flow.

The heading of background literature review, “Part one: Background literature review on purchasing behaviours of sweetened beverages” (Line 97-98). A summary table has been added to the beginning of this section to provide an overview of individual and contextual factors on sweetened beverage behaviour (Lines 117-120).

The heading of the audit study has been amended as “Part two: Cross-sectional audit study”. (Line 238).

Lines 99-103: Please include more details on the results of the literature search.

Author response: More information has now been added, “A keyword search (“grab and go”, “ready to go”, “sugar-sweetened beverages”, “consumer behaviour”, “serving size”, ‘portion size’) was conducted in Medline via Ovid, Passport Euromonitor and IBISWorld (from inception to June 2023), as these are the most relevant health and marketing databases. The search yielded 107 research articles and reports.  Backward and forward hand searching was also undertaken using relevant studies from the preliminary search and yielding another 18 articles [4-6].” (Lines 99-104).

L 123-125: Please write in the text that this applies to Australia.

Author response: This has been amended as “The vast majority of soft drinks are sold through supermarkets as confirmed by Australian industry reports.” (Lines 127-128).

Discussion:

The first paragraph is actually a summary of the results, consider moving or rephrasing this paragraph.

Author response: Thank you for your suggestion, this paragraph has been amended:

“This study provides a novel insight into the variety of sweetened beverages available in the GNG food environment across three socioeconomic localities within Sydney. Over 1200 sweetened beverages were included, consisting of a wide range of serving sizes and energy values. The majority of all GNG beverages were sugar-sweetened and approximately two-thirds exceeded the recommended medium serving size of 375 mL set by the Healthy Food Partnership in The Industry Guide to Voluntary Serving Size Reduction [7]. No differences were observed between the socioeconomic localities in terms of proportion of sugar-sweetened beverages, serving sizes or energy values.” (Lines 379-386).

Conclusions:

L439-440: Add “in Australia”.

Author response: This has now been added, “This study provides a novel insight into the availability of sweetened discretionary drinks in the relatively new GNG food environment in Australia.” (Line 437-438).

L444-447- (450): Move to Discussion. This has not been discussed previously, and new information should not be presented in Conclusions. Also, references should not be included in Conclusions.

Author response: The Conclusion section has been amended as per suggestions,

“This study provides a novel insight into the availability of sweetened discretionary drinks in the relatively new GNG food environment in Australia. The majority of GNG beverages had serving sizes exceeding one standard discretionary serve (600 kJ), resulting from large serving sizes >375 mL making such sizes the ‘norm’ in the GNG section. These findings highlight the need to create food environments that enable consumers to choose more appropriate portion sizes, compatible with their energy needs. Active engagement of stakeholders including the food industry, retailers and government bodies, together with clear guidelines around recommended serving sizes, is critical to assist consumers to select appropriate serving sizes and avoid overconsumption.” (Lines 437-445).

References

  1. Hollands, G.J.; Shemilt, I.; Marteau, T.M.; Jebb, S.A.; Lewis, H.B.; Wei, Y.; Higgins, J.P.; Ogilvie, D. Portion, package or tableware size for changing selection and consumption of food, alcohol and tobacco. Cochrane Database Syst Rev 2015, 2015, Cd011045, doi:10.1002/14651858.CD011045.pub2.
  2. Zlatevska, N.; Dubelaar, C.; Holden, S.S. Sizing up the Effect of Portion Size on Consumption: A Meta-Analytic Review. J Mark 2014, 78, 140-154, doi:10.1509/jm.12.0303.
  3. Australian Bureau of Satistics. Socio-Economic Indexes for Areas (SEIFA), Australia. Availabe online: https://www.abs.gov.au/statistics/people/people-and-communities/socio-economic-indexes-areas-seifa-australia/2021?#media-releaseshttps://www.smh.com.au/national/nsw/sydney-s-richest-and-poorest-postcodes-how-does-your-area-compare-20230426-p5d3bd.html (accessed on 30 June 2023).
  4. Caspi, C.E.; Sorensen, G.; Subramanian, S.V.; Kawachi, I. The local food environment and diet: a systematic review. Health Place 2012, 18, 1172-1187, doi:10.1016/j.healthplace.2012.05.006.
  5. Dono, J.; Ettridge, K.; Wakefield, M.; Pettigrew, S.; Coveney, J.; Roder, D.; Durkin, S.; Wittert, G.; Martin, J.; Miller, C. Nothing beats taste or convenience: a national survey of where and why people buy sugary drinks in Australia. Aust N Z J Public Health 2020, 44, 291-294, doi:10.1111/1753-6405.13000.
  6. Purohit, B.M.; Dawar, A.; Bansal, K.; Nilima; Malhotra, S.; Mathur, V.P.; Duggal, R. Sugar-sweetened beverage consumption and socioeconomic status: A systematic review and meta-analysis. Nutr Health 2022, 10.1177/02601060221139588, 2601060221139588, doi:10.1177/02601060221139588.
  7. Australian Department of Health. Healthy Food Partnership: Industry Guide to Voluntary Serving Size Reduction; 2023.

This manuscript is a resubmission of an earlier submission. The following is a list of the peer review reports and author responses from that submission.

Round 1

Reviewer 1 Report

In my opinion, this study titled " Serving sizes and energy content of grab-and-go beverages in convenience stores, supermarkets, and fast-food outlets " is mainly a relevant study on the response of consumer behavior to beverage choices in three different retail venues. It is suggested that the authors can cite the theoretical basis to strengthen the overall structure of the article. In addition, the literature review's series rigor and research method design also need to be strengthened. Furthermore, whether differences in the behavior or choices of research subjects or research samples result in retailers or the beverage industry can provide research contributions. The discussion in the conclusion is too weak, and it is recommended to strengthen it. Finally, the conclusions and recommendations also need to echo the purpose and importance of the research to strengthen logical consistency.

Author Response

Reviewer 1
In my opinion, this study titled " Serving sizes and energy content of grab-and-go beverages in convenience stores, supermarkets, and fast-food outlets " is mainly a relevant study on the response of consumer behavior to beverage choices in three different retail venues.

Author response: Thank you for your comments and review of this manuscript.

It is suggested that the authors can cite the theoretical basis to strengthen the overall structure of the article.

Author response:

The external food environment plays a key role in portion control and eating behaviours as the portion size effect has been consistently demonstrated (that is, exposing consumers to larger serving sizes leads to unconscious overconsumption) [1, 2]. The associated negative health consequences (for example, dental caries, overweight, hypertension, diabetes) of excessive sweetened beverages consumption have also been well documented [3, 4]. Therefore, this study aimed to collect data on GNG sweetened beverages to better understand this relatively new food environment.

These theories have now been added to the Introduction section, “In Australia, the latest national nutrition survey showed one third of the population consumed a median of one can of sugar-sweetened beverages per day [5]. Excessive consumption of such beverages has been associated with dental caries, overweight, obesity, hypertension, diabetes and cardiovascular disease [3, 4].” (Lines 35-39);

“It is well established that exposing consumers to larger serving sizes leads to unconscious overconsumption [6, 7]. In addition, the current widespread availability of food and drinks in large serving sizes has shifted the perceived norm of portion sizes to be larger than previous [6, 7]. As larger sizes are becoming the new normal, accompanied by higher energy intakes, the risk of overweight, obesity and other chronic disease increases in the population [6, 7].

Given the key role of the external food environment on eating behaviours [1, 2], a better understanding of the GNG food environment would help identify potential areas for public health interventions to improve health outcomes.” (Lines 61-69).

In addition, the literature review's series rigor and research method design also need to be strengthened.

Author response: The literature review was conducted to better describe the contextual background around consumer purchasing behaviours of sweetened beverages. Searches were conducted in Medline via Ovid, Passport Euromonitor, and IBISWorld, as well as forward and backward hand search using relevant studies identified from the preliminary search [8-10]. This information has been now added to the literature review section, “A keyword search (for example, “grab and go”, “ready to go”, “sugar-sweetened beverages”, “consumer behaviour”) was conducted in health and marketing databases including Medline via Ovid, Passport Euromonitor and IBISWorld. Backward and forward hand searching was undertaken using relevant studies from the preliminary search [8-10]. (Lines 77-81).

Furthermore, whether differences in the behavior or choices of research subjects or research samples result in retailers or the beverage industry can provide research contributions.

Author response: This study aimed to emphasise the high availability of sweetened beverages in large serving sizes and compare current serving sizes with the recommendations [11]. Such data could help understand this relatively new GNG food environment and highlight potential areas for the development of future public health interventions and public health advocacy. The potential implications have now been acknowledged in the introduction and conclusion sections:

“It is well established that exposing consumers to larger serving sizes leads to unconscious overconsumption [6, 7]. In addition, the current widespread availability of food and drinks in large serving sizes has shifted the perceived norm of portion sizes to be larger than previous [6, 7]. As larger sizes are becoming the new normal, accompanied by higher energy intakes, the risk of overweight, obesity and other chronic disease increases in the population [6, 7].

Given the key role of the external food environment on eating behaviours [1, 2], a better understanding of the GNG food environment would help identify potential areas for public health interventions to improve health outcomes.” (Lines 61-69).

“Sweetened discretionary drinks in large sizes are likely to be the ‘norm’ in the GNG section. These findings highlight the need to create food environments that enable consumers to choose more appropriate portion sizes, compatible with their needs. Multiple public health strategies have been proposed to reduce sugary beverages consumption such as sugar taxes, menu labelling, replacing soft drinks with healthier alternatives such as milk-based drinks, and setting cap rules on single serve beverages [12, 13]. Increasing the availability of smaller serving size options at point-of-purchase has been recommended as one of the most promising and acceptable strategies to nudge consumers towards more appropriate serving sizes [14-16]. Active engagement of stakeholders including the food industry, retailers and government bodies, together with clear guidelines around recommended serving sizes, is critical to assist consumers to select appropriate serving sizes and avoid overconsumption.” (Lines 400-411).

The discussion in the conclusion is too weak, and it is recommended to strengthen it.

Finally, the conclusions and recommendations also need to echo the purpose and importance of the research to strengthen logical consistency.

Author response: The conclusion section has been amended, “The majority of GNG beverages had serving sizes exceeding one standard discretionary serve (600 kJ), resulting from a large serving size  >375 mL. Sweetened discretionary drinks in large sizes are likely to be the ‘norm’ in the GNG section. These findings highlight the need to create food environments that enable consumers to choose more appropriate portion sizes, compatible with their needs. Multiple public health strategies have been proposed to reduce sugary beverages consumption such as sugar taxes, menu labelling, replacing soft drinks with healthier alternatives such as milk-based drinks, and setting cap rules on single serve beverages [12, 13]. Increasing the availability of smaller serving size options at point-of-purchase has been recommended as one of the most promising and acceptable strategies to nudge consumers towards more appropriate serving sizes [14-16]. Active engagement of stakeholders including the food industry, retailers and government bodies, together with clear guidelines around recommended serving sizes, is critical to assist consumers to select appropriate serving sizes and avoid overconsumption.” (Lines 398-411).

References

  1. Hollands GJ, Shemilt I, Marteau TM, Jebb SA, Lewis HB, Wei Y, et al. Portion, package or tableware size for changing selection and consumption of food, alcohol and tobacco. Cochrane Database Syst Rev. 2015;2015(9):Cd011045.
  2. Zlatevska N, Dubelaar C, Holden SS. Sizing up the Effect of Portion Size on Consumption: A Meta-Analytic Review. Journal of Marketing. 2014;78(3):140-154.
  3. Bleich SN, Vercammen KA. The negative impact of sugar-sweetened beverages on children’s health: an update of the literature. BMC Obesity. 2018;5(1):6.
  4. Li B, Yan N, Jiang H, Cui M, Wu M, Wang L, et al. Consumption of sugar sweetened beverages, artificially sweetened beverages and fruit juices and risk of type 2 diabetes, hypertension, cardiovascular disease, and mortality: A meta-analysis. Front Nutr. 2023;10:1019534.
  5. Australian Bureau of Satistics. Australian Health Survey: Nutrition First Results - Foods and Nutrients 2019 [Available from: https://www.abs.gov.au/statistics/health/health-conditions-and-risks/australian-health-survey-nutrition-first-results-foods-and-nutrients/latest-release#discretionary-foods.
  6. Mattes RD. Evidence on the “normalizing” effect of reducing food-portion sizes. The American Journal of Clinical Nutrition. 2018;107(4):501-503.
  7. Steenhuis IHM, Vermeer WM. Portion size: review and framework for interventions. International Journal of Behavioral Nutrition and Physical Activity. 2009;6(1):58.
  8. Caspi CE, Sorensen G, Subramanian SV, Kawachi I. The local food environment and diet: a systematic review. Health Place. 2012;18(5):1172-1187.
  9. Dono J, Ettridge K, Wakefield M, Pettigrew S, Coveney J, Roder D, et al. Nothing beats taste or convenience: a national survey of where and why people buy sugary drinks in Australia. Aust N Z J Public Health. 2020;44(4):291-294.
  10. Purohit BM, Dawar A, Bansal K, Nilima, Malhotra S, Mathur VP, et al. Sugar-sweetened beverage consumption and socioeconomic status: A systematic review and meta-analysis. Nutr Health. 2022:2601060221139588.
  11. Australian Department of Health. Healthy Food Partnership: Industry Guide to Voluntary Serving Size Reduction. 2023.
  12. Vercammen KA, Frelier JM, Lowery CM, Moran AJ, Bleich SN. Strategies to reduce sugar-sweetened beverage consumption and increase water access and intake among young children: perspectives from expert stakeholders. Public Health Nutr. 2018;21(18):3440-3449.
  13. von Philipsborn P, Stratil JM, Burns J, Busert LK, Pfadenhauer LM, Polus S, et al. Environmental interventions to reduce the consumption of sugar-sweetened beverages and their effects on health. Cochrane Database Syst Rev. 2019;6(6):Cd012292.
  14. Hetherington MM, Blundell‐Birtill P. The portion size effect and overconsumption – towards downsizing solutions for children and adolescents. Nutrition bulletin. 2018;43(1):61-68.
  15. Liu Q, Tam LY, Rangan A. The Effect of Downsizing Packages of Energy-Dense, Nutrient-Poor Snacks and Drinks on Consumption, Intentions, and Perceptions-A Scoping Review. Nutrients. 2021;14(1).
  16. Steenhuis I, Poelman M. Portion Size: Latest Developments and Interventions. Curr Obes Rep. 2017;6(1):10-17.

Reviewer 2 Report

The authors present a survey study on grab-and-go (GNG) beverages, located in the metropolitan area of the city of Sydney.

Paper writing and conceptualization are in line with the scope of the journal. Bibliographic sources throughout the paper appear adequate; however, the structuring of the paper as a survey based on the study of bibliographic sources specifically concerning sweetened beverages, although useful for carrying out the survey, could mislead the reader about the consistency of the survey methodology. It is therefore advisable to specify the main focus on sweetened beverages in the title of the paper.

Scientific content of the paper shows a rather simple methodological approach, with an emphasis on descriptive statistics integrated with the use of methods for verifying the equality of the medians of different groups.

As regards presentation of results, in some places it is necessary to better specify in the text some parameters indicated in the figures, such as the "standard discretionary serve" shown in Figure 1, the definition of which should be integrated at the bottom of the comment in the text [after line 283].

In discussing the results and in the subsequent conclusions, the authors make no secret of some evident methodological limitations, such as for example the survey conducted only on available products, or the choice of a city survey area. To these limitations should certainly be added the fact of having analyzed mainly sweetened drinks, an aspect that should be integrated in the paragraph "strengths and limitation" or in the conclusions.

From the above indications and comments in the opinion of this reviewer the paper, apart from the metholodlogy shortcomings, is suitable for publication after the modifications suggested.

Author Response

Reviewer 2

The authors present a survey study on grab-and-go (GNG) beverages, located in the metropolitan area of the city of Sydney.

Author response: Thank you for your comments and review of this manuscript.

Paper writing and conceptualization are in line with the scope of the journal. Bibliographic sources throughout the paper appear adequate; however, the structuring of the paper as a survey based on the study of bibliographic sources specifically concerning sweetened beverages, although useful for carrying out the survey, could mislead the reader about the consistency of the survey methodology. It is therefore advisable to specify the main focus on sweetened beverages in the title of the paper.

Author response: Thank you for your suggestions. The title has been amended as “Serving sizes and energy content of grab-and-go sweetened beverages in convenience stores, supermarkets, and fast-food outlets”.

Scientific content of the paper shows a rather simple methodological approach, with an emphasis on descriptive statistics integrated with the use of methods for verifying the equality of the medians of different groups.

Author response: This has now been acknowledged in the data analysis section: “. Non-parametric tests were used due to data constituting ordinal values.“ (Lines 246-247).
As regards presentation of results, in some places it is necessary to better specify in the text some parameters indicated in the figures, such as the "standard discretionary serve" shown in Figure 1, the definition of which should be integrated at the bottom of the comment in the text [after line 283].

Author response: The distribution of small, medium, and large sugar-sweetened GNG beverages serving sizes in three settings were presented in Figure 1. This information has now been added for better clarity, “The percentages of small, medium, and large sugar-sweetened GNG beverages in three settings are presented in Figure 1. Beverages were classified as small if serving size <250 mL, medium if 251-375 mL, large if >375 mL; one standard discretionary serve equals 600 kJ as per the Australian Dietary Guidelines [1-3].” (Lines 289-291).

In discussing the results and in the subsequent conclusions, the authors make no secret of some evident methodological limitations, such as for example the survey conducted only on available products, or the choice of a city survey area. To these limitations should certainly be added the fact of having analyzed mainly sweetened drinks, an aspect that should be integrated in the paragraph "strengths and limitation" or in the conclusions.

Author response: This has now been acknowledged in this limitation section, “This study focused on sweetened beverages at the most common GNG settings only, other GNG beverage types (for example, flavoured milks, mineral drinks, water) and those available in smaller retail outlets (for example, milk bars and corner stores) were not collected. (Lines 387-391).

From the above indications and comments in the opinion of this reviewer the paper, apart from the methodology shortcomings, is suitable for publication after the modifications suggested.

Author response: Thank you for your suggestions and review of this manuscript.

References

  1. National Health and Medical Research Council. Australian Dietary Guidelines. In: Ageing DoHa, editor. Canberra: National Health and Medical Research Council; 2013.
  2. Chepulis L, Mearns G, Hill S, Wu JHY, Crino M, Alderton S, et al. The nutritional content of supermarket beverages: a cross-sectional analysis of New Zealand, Australia, Canada and the UK. Public Health Nutrition. 2018;21(13):2507-2516.
  3. Poelman MP, Eyles H, Dunford E, Schermel A, L’Abbe MR, Neal B, et al. Package size and manufacturer-recommended serving size of sweet beverages: a cross-sectional study across four high-income countries. Public Health Nutrition. 2016;19(6):1008-1016.
